# Chitosan-Based Delivery Systems Loaded with Glibenclamide and Lipoic Acid: Formulation, Characterization, and Kinetic Release Studies

**Luminita-Georgeta Confederat [1], Iuliana Motrescu [2], Mihaela Iustina Condurache [3], Sandra Constantin [4], Alexandra Bujor [5], Cristina Gabriela Tuchilus [1] and Lenuta Profire [4,*]**

1   Department of Microbiology, "Grigore T. Popa" University of Medicine and Pharmacy of Iasi, 700115 Iasi, Romania; georgeta-luminita.confederat@umfiasi.ro (L.-G.C.); cristina.tuchilus@umfiasi.ro (C.G.T.)
2   Department of Exact Sciences, "Ion Ionescu de la Brad" University of Agricultural Sciences and Veterinary Medicine, 700490 Iași, Romania; imotrescu@uaiasi.ro
3   Department of Biomedical Sciences, "Grigore T. Popa" University of Medicine and Pharmacyof Iasi, 700115 Iasi, Romania; mihaela-iustina.condurache@umfiasi.ro
4   Department of Pharmaceutical Chemistry, "Grigore T. Popa" University of Medicine and Pharmacy of Iasi, 700115 Iasi, Romania; constantin.sandra@umfiasi.ro
5   Department of Pharmaceutical Technology, "Grigore T. Popa" University of Medicine and Pharmacy of Iasi, 700115 Iasi, Romania; alexandra.m.bujor@umfiasi.ro
*   Correspondence: lenuta.profire@umfiasi.ro



**Featured Application: This article presents the development and characterization of new chitosan based polymeric systems containing glibenclamide and lipoic acid for diabetes mellitus therapy. The systems were designed as multi-target systems, aiming for optimal glycemic control, as well as the reactive oxygen species generated by hyperglycemia.**

**Abstract:** Glibenclamide and lipoic acid are two drugs frequently recommended for the management of diabetes mellitus, and so, the development of a new formulation containing both substances has a great benefit in terms of efficiency and compliance, acting also as a multi-target drug system. Accordingly, the aim of this study was the formulation and physicochemicalcharacterization of new polymeric systems based on chitosan (CS) in whose matrix were encapsulated glibenclamide (Gly) and lipoic acid (LA). The polymeric systems were prepared as microparticles (CS–Gly, CS–LA, and CS–Gly–LA) through ionic gelation method, using pentasodium tripolyphosphate (TPP) as crosslinking agent. The polymeric systems obtained were characterized in terms of particle size and morphology, IR spectroscopy, entrapment efficiency and drug loading, swelling degree, and therelease of the active substances from the chitosan matrix. The polymeric systems obtained were stable systems; the presence of glibenclamide and lipoic acid into the polymer matrix were proved by IR spectroscopy. The entrapment efficiency was 94.66% for Gly and 39.68% for LA. The developed polymeric systems proved a favorable swelling degree and drug release profile, the percentage of release being 88.68% for LA and 75.17% for Gly from CS–Gly–LA systems.

**Keywords:** chitosan; glibenclamide; lipoic acid; drug delivery systems

## 1. Introduction

Diabetes mellitus (DM) is a major public health problem that represents, nowadays, an important concern due to its increasing prevalence and due to the challenges related to itstreatment, which is

one of the major causes of mortality [1]. The specific symptom of this chronic metabolic disorder is hyperglycemia due to a deficiency in insulin secretion, caused by damaged pancreatic beta-cells, or, peripherally, insulin resistance, or both of them [2].

DM is associated with severe micro- and macro vascular complications, such as retinopathy [3,4], neuropathy, nephropathy [5,6], diabetic foot [7], atherosclerosis, myocardial infarction and stroke [8,9], rheumatoid arthritis, renal failure, and neurodegenerative disorders [6].

Conventional antidiabetic drugs include sulfonylureas, biguanides, thiazolidinediones, alpha-glucosidase inhibitors, glinides, andmodulators of incretins (glucagon-like peptide-1 receptor-agonists and dipeptidyl peptidase IV enzyme inhibitors) [10,11]. The major target of antidiabetic therapy is to reduce the increased blood glucose level, which, in several cases, is very difficult to achieve [12]. In addition, many antidiabetic drugs are responsible for different side effects, such as hypoglycemia, liver and kidney injury, weight gain, gastrointestinal disturbances, and hypersensitivity reactions [13,14].

Glibenclamide (Gly) is a second-generation sulfonylurea, which is frequently prescribed for diabetes mellitus type 2 (DM2), when the treatment with biguanides and first-generation sulfonylureas cannot ensurean optimal glycemic control [13,14]. It acts mainly bystimulating insulin secretion in pancreatic beta-cells and, consequently, by reducingthe blood glucose levels [9,10]. Despite its intense and long-term action, Gly has poor aqueous solubility, which causes reduced and variable oral bioavailability [15,16].

The chronic hyperglycemia, which is the major cause of diabetes mellitus complications, promotes, also, oxidative stress through free reactive oxygen species (ROS) generation and suppression of antioxidant defense system [17,18]. In addition, there is some evidence thatinvolves ROS in impaired beta-cells function caused by proinflammatory cytokines and autoimmune reactions [19]. Based on the implication of oxidative stress in the progression of diabetes mellitus and its complications, several antioxidants were evaluated for their beneficial effects [20]. Common antioxidants include vitamins (A, C, and E), glutathione, enzymes (superoxide dismutase, catalase, glutathione reductase), lipoic acid, coenzyme Q10, cofactors (folic acid, vitamins B1, B2, B6, B12), and minerals (copper, zinc, selenium, and manganese) [18,21].

For lipoic acid (LA), literature data reveals important pharmacological effects, such as antioxidant [21,22], antitumor and anti-inflammatory activity [23], as well as increasing the insulin sensitivity [24]. Moreover, LA is frequently used for relieving the diabetic neuropathy symptoms [25].

This, despite there are available different classes of oral antidiabetic drugs, with different mechanisms of action—new, safer, and effective agents are needed.Moreover, in recent years, many studies focused on the development of drug delivery systems with controlled or sustained release, in order to improve the pharmacokinetic profile of conventional drugs or to assure a safety profile [26,27].

Chitosan (CS) is a natural biopolymer obtained by alkaline deacetylation of chitin, whose chemical structure consists in β-1,4-linked D-glucosamine and N-acetyl-D-glucosamine units [28]. It is biodegradable, biocompatible, non-toxic, has low immunogenicity [28,29], and could be derivatized at the amino groups [30]. In addition, CS is known to have significant biological effects such as antidiabetic, antioxidant, antitumor, antimicrobial, anti-inflammatory, cholesterol lowering properties [30,31]. Due to its physicochemical and biological features, CS is widely investigated for pharmaceutical and biomedical applications [32], including drug delivery systems [26–28].

Based on the biological effects of Gly and LA, a new CS-based formulation, which contains both substances, was designed as multi-target drug system, acting mainly as astimulator of insulin secretion in pancreatic beta-cells, as well as free reactive oxygen species scavenger. Therefore, in this study the preparation, physicochemical characterization, and kinetic drug release studies of new polymeric systems based on CS, Gly, and LA are presented.

## 2. Materials and Methods

### 2.1. Materials

Chitosan (CS, medium molecular weight 190–310 kDa, degree of deacetylation 75–85%), pentasodium tripolyphosphate (TPP, ≥85%), acetic acid (≥ 99.8%), methanol (≥ 99.8%), glibenclamide (Gly, ≥ 99%) and lipoic acid (LA, ≥ 98%) were purchased from Sigma Aldrich (Merck, Germany). Distilled water of chromatographic purity was purchased from Ircon (Romania). Taurocholic acid sodium salt (≥ 95%), lecithin (≥ 60%), pepsin obtained from gastric porcine mucosa (≥ 250 units/mg powder), sodium chloride (NaCl; ≥ 99%), hydrochloric acid (HCl; 37%), maleic acid (≥ 99%) and sodium hydroxide (≥97%) were also purchased from Sigma Aldrich (Merck, Germany).

### 2.2. Preparation of the Polymeric Systems

The new polymeric systems, based on CS microparticles loaded with Gly and LA (CS–Gly, CS–LA, and CS–Gly–LA), were obtained through ionic gelation technique, which is based on the electrostatic interactions between the positively charged amino groups of CS and the negatively charged groups of TPP [29]. CS solution, in concentration of 1.5%, was prepared by dissolving the appropriate amount of CS in 100 mL acetic acid solution 1% *v/v*, under stirring. Separately, TPP solution, in concentration of 2% was prepared by dissolving the appropriate amount of TPP in distilled water [33]. In order to obtain stable polymeric systems, different ratios of CS:Gly and CS:LA were used (1:1, 1:0.75, 1:0.5). The best ratio was 1:1 for both CS–Gly and CS–LA systems. For CS–Gly–LA polymeric system the optimum ratio was 1:0.5:0.5. The active substances (Gly, LA) in the established ratio, were dissolved in the minimum amount of methanol and then were added into 3 mL of CS solution 1.5%. The mixture was stirred for 2 h, at 350 rpm; after that, the microparticles were obtained by adding, dropwise, 3 mL of the mixture obtained into 20 mL of TPP solution, using a syringe of 1 mL, with a 27-gauge needle, the drop's volume being 6.25 µL, under stirring (250 rpm) [30]. The dropping rate was 15 drops/min and the falling distance was approximatively 10 cm. An additional stirring (200 rpm) for 5h was realized in order to increase the stability of the formed microparticles; then, the microparticles obtained were separated, washed with distilled water, and dried at room temperature.

### 2.3. Characterization of the Polymeric Systems

#### 2.3.1. Particle Size and Morphology

The size of the CS, CS–Gly, CS–LA, and CS–Gly–LA microparticles, in wet state, was determined using an optical microscope model Leica DM 750 with a built-in video camera, model ICC50 W0366, the images being recorded with 10× objective.

The surface morphology and the size of the microparticles in dry state were analyzed by Environmental Scanning Electron Microscopy (ESEM) using a Quanta 450 (Thermo Fisher Scientific, Hillsboro, Oregon, USA) microscope. The ESEM microanalysis can be performed for any samples, even non-conductive without the necessity of applying a conductive layer that can influence the surface topography. Samples were placed on an aluminum stub covered with a carbon layer, and images were achieved by applying an electron beam with an accelerating voltage of 20 kV. The diameters of the microparticles were calculated using ESEM software. All of the measurements were performed in triplicate and the values were expressed as mean ± standard deviation (SD).

#### 2.3.2. Fourier Transform Infrared Spectroscopy (FT-IR)

The IR spectra of CS, active substances (Gly, LA) and polymeric systems (CS–Gly, CS–LA and CS–Gly–LA) were recorded using a Fourier transform infrared spectrophotometer model ABB-MB3000 FT-IR MIRacle TM Single Bounce ATR (Attenuated Total Reflectance), operating from 4000 to 650 cm$^{-1}$, at a resolution of 4 cm$^{-1}$. The IR spectra were processed after 16 successive measurements using Horizon MBTM FT-IR Software.

### 2.3.3. Yield, Entrapment Efficiency, and Drug Loading

The yield ($\eta$%), representing the percentage of CS and active substances (Gly, LA), which was transformed/loaded into microparticles, was calculated using the formula [34]:

$$\eta(\%) = (m_1/m_0) \times 100 \qquad (1)$$

where:

$m_1$ = the amount of obtained microparticles, in dry state (mg)
$m_0$ = the total initial amount of CS and active substances (Gly, LA) (mg)

The entrapment efficiency (EE%) was calculated based on a validated High Performance Liquid Chromatography (HPLC) method, using a HPLC apparatus type Agilent Technologies 1200 Series, equipped with UV detector type DAD (Diode Array Detector) G1315A and injector type Rheodyne. The following parameters were used: acetonitrile and phosphate buffer, pH = 2.7, as mobile phase, gradient elution during 60 min, at a flow rate of 1 mL/min [35,36], ODS (octadecyl-silica) Hypersil C18 column, 250 × 4.6 mm, 5 μm particles size. Briefly, after removing the formed CS–Gly, CS–LA, and CS–Gly–LA microparticles, the TPP solution was analyzed in the mentioned chromatographic conditions. Based on the peak area corresponding to Gly and LA and on the corresponding specific calibration curves, the unloaded amount of active substances (Gly, LA) was determined. By difference between the initial ($m_1$) and unloaded amount ($m_2$) of active substances, the amount of loaded ($m_3$) active substances into the chitosan matrix was calculated. EE% was calculated using the formula [34]:

$$EE\% = (m_3/m_0) \times 100 \qquad (2)$$

where:

$m_3$ = the amount of active substance loaded into CS microparticles (mg)
$m_0$ = the initial amount of active substance used for the preparation of microparticles (mg)

Drug loading (DL%) was calculated using the following formula [37]:

$$DL\% = (m_3/m_1) \times 100 \qquad (3)$$

where:

$m_3$ = the amount of active substance loaded into chitosan microparticles (mg)
$m_1$ = the amount of microparticles obtained, in dry state (mg)

All tests were performed in triplicate and the results were expressed as mean ± SD.

### 2.3.4. Swelling Degree

Swelling degree (SD%) was evaluated for CS, CS–Gly, CS–LA, and CS–Gly–LA polymeric systems using distilled water and simulated gastric fluid (SGF), pH = 1.6. The experiments were performed according to literature data with slight modifications [38,39]. Briefly, an exactly amount of microparticles was introduced in distilled water and in SGF. At different timelines, the microparticles were removed from the media, dried quickly with filter paper, and weighted in wet state ($w_1$). Then microparticles were reintroduced in the media and the operation was repeated up to constant weight. At the end of the experiment, the microparticles (CS, CS–Gly, CS–LA, CS–Gly–LA) were removed, dried at room temperature, and weighted in dried state ($w_2$). For distilled water, the determinations were realized during 24 h and for SGF during 2 h, considering the physiological passage time in the gastrointestinal tract. The tests were performed in triplicate and the results were expressed as mean ± standard deviation. SD% was calculated gravimetrically using the formula:

$$SD\% = [(w_1 - w_2)/w_2] \times 100 \qquad (4)$$

where:

$w_1$ = the weight of the swollen microparticles at different timelines (mg)
$w_2$ = the weight of the dried microparticles (mg)

### 2.3.5. KineticDrug Release Studies

The release of the active substances (Gly, LA) from the polymeric systems, CS–Gly, CS–LA, and CS–Gly–LA, was studied according to the methods described in the literature with slight modifications [40–42]. Simulated gastric (SGF, pH = 1.6, sodium taurocholate 0.08 mM, lecithin 0.02 mM, pepsin 0.1 mg/mL, sodium chloride 34.2 mM and hydrochloric acid 25.1 mM) and intestinal (SIF (simulated intestinal fluid), pH = 6.5, sodium taurocholate 3 mM, lecithin 0.2 mM, maleic acid 19.12 mM, sodium hydroxide 34.80 mM, and sodium chloride 68.62 mM) fluids were used [43].

A weighted amount of CS–Gly, CS–LA, and CS–Gly–LA microparticles, was introduced in 1.8 mL simulated media (SGF, SIF) and kept at 37 ± 1 °C, under stirring. At constant timelines (at each 60 min during 2 h for SGF and at each 60 min for 8 h for SIF), 0.5 mL of simulated media was extracted and analyzed in the chromatographic conditions mentioned above, using a HPLC apparatus type Agilent Technologies 1200 Series, equipped with UV detector type DAD G1315A and injector type Rheodyne. Based on the peak area corresponding to Gly and LA and on the corresponding specific calibration curve, the amount of active substances (Gly, LA), released from polymer matrix at each time interval was determined ($m_t$). The degree of release (DR%) was calculated using the following formula:

$$DR\% = (m_t/m_0) \times 100 \tag{5}$$

where:

$m_0$ = the initial amount of active substance (Gly, LA) loaded into microparticles (mg)
$m_t$ = the amount of active substance released, at a specific time, in simulated media (mg)

All of the experiments were performed in triplicate and the results were expressed as mean ± SD.

## 3. Results

### 3.1. Particle Size and Morphology

The size of the CS microparticles were 798.15 ± 18.60 μm in wet state and 522.30 ± 16.26 μm in dry state. The loading of the active substances was associated with increasing of the particles size. As a result, the size of CS–Gly microparticles was 822.00 ± 23.11 μm in wet state and 532.98 ± 40.08 μm in dry state, while the size of CS–LA microparticles was 1130.00 ± 31.25 μm in wet state and 916.73 ± 39.91 μm in dry state. For CS–Gly–LA, the size of microparticles ranged from 1079.00 ± 25.48 μm in wet state to 863.74 ± 32.19 μm in dry state. SEM images of the CS, CS–Gly, CS–LA, and CS–Gly–LA are shown in Figure 1. It was noted that CS microparticles (Figure 1A) are spherical, with regular shape, and with a well-organized polymeric network. Encapsulation of active substances (Gly, LA) into CS matrix was associated with a slight (for CS–Gly, Figure 1B)) and intense (CS–LA, Figure 1C)) shape deformation. For these polymeric systems a slightlyrough surface was also observed. For the new polymeric systems, CS–Gly–LA, a spherical shape with a slightly rough surface was also noted (Figure 1D).

### 3.2. Fourier Transform Infrared Spectroscopy (FT-IR)

The analysis of the IR spectra of the developed polymeric systems (Figure 2) proved the encapsulation of the active substances into chitosan matrix through the presence of the characteristic absorption bands for CS, Gly, and LA. The spectra of CS and CS microparticles revealed the peak associated to the vibration of carbonyl bond of amide groups (-CO-NH-) at 1660 cm$^{-1}$ and a peak at 1589 cm$^{-1}$ corresponding to the vibration of amine groups (NH$_2$). The peaks at 1430 cm$^{-1}$

and 1060 cm$^{-1}$ were attributed to the vibrations of methylene groups (CH$_2$) and etheric bond (C-O-C), respectively. In addition, the large band between 3100–3600 cm$^{-1}$ was attributed to the vibration of -OH and -NH- groups.

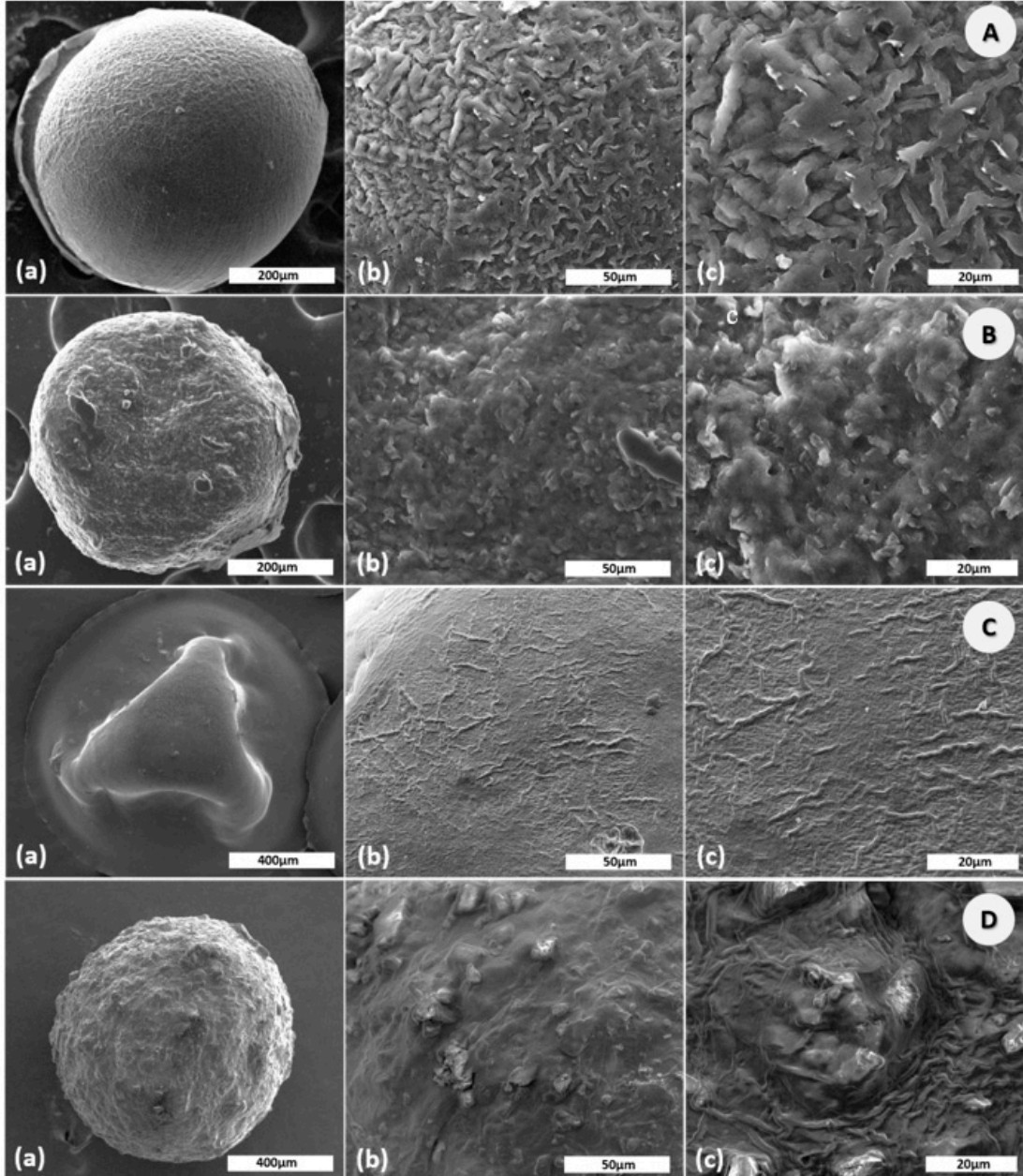

**Figure 1.** SEM images ofchitosan (CS) (**A**); CS-glibenclamide (Gly) (**B**), CS-lipoic acid (LA) (**C**), CS–Gly–LA (**D**) microparticles at different scales: 200/400 µm (a), 50 µm (b), 20 µm (c).

The presence of Gly into CS–Gly–LA microparticles was proved by the absorption characteristic bands corresponding to stretching vibrations for C-H groups from the aromatic ring (2950–2850 cm$^{-1}$), carbonyl group C=O (1690 cm$^{-1}$), deformation vibrations for secondary amino group NH- (1400–1460 cm$^{-1}$), S=O (1350–1300 cm$^{-1}$) and 1150 cm$^{-1}$ (C-N). LA was proved into CS–Gly–LA microparticles by the presence of the absorption characteristic bands corresponding to methylene groups CH$_2$ (2950–2900 cm$^{-1}$), carbonyl C=O (1690 cm$^{-1}$), C-H (1430–1380 cm$^{-1}$), hydroxyl OH (950 cm$^{-1}$), C-S (750–700 cm$^{-1}$), S-S (671 cm$^{-1}$) groups.

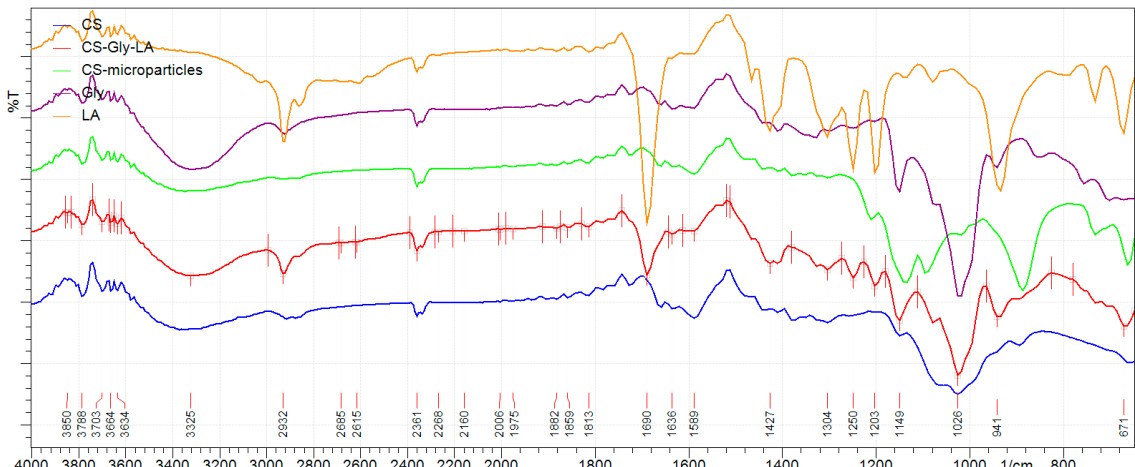

**Figure 2.** IR spectra of glibenclamide (Gly, pure substance), lipoic acid (LA, pure substance), chitosan (CS, pure substance), CS microparticles and CS–Gly–LA microparticles.

### 3.3. Yield, Entrapment Efficiency, and Drug Loading

The yield preparation of CS–Gly, CS–LA, and CS–Gly–LA microparticles in the ratio of 1:1, 1:1, and 1:0.5:0.5, expressed as the percentage of substances (CS, Gly, LA) transformed/loaded into microparticles, was 98.5% for CS–Gly microparticles, 63.22% for CS–LA microparticles and 78.33% for CS–Gly–LA microparticles, respectively.

The standard curves for Gly and LA were linear in the established concentration range and the correlation coefficient ($R^2$) was within the specified limits ($R^2 > 0.98$; 0.9992 for Gly and 0.9994 for LA, respectively), which proves the reproducibility of the method (Figure 3).

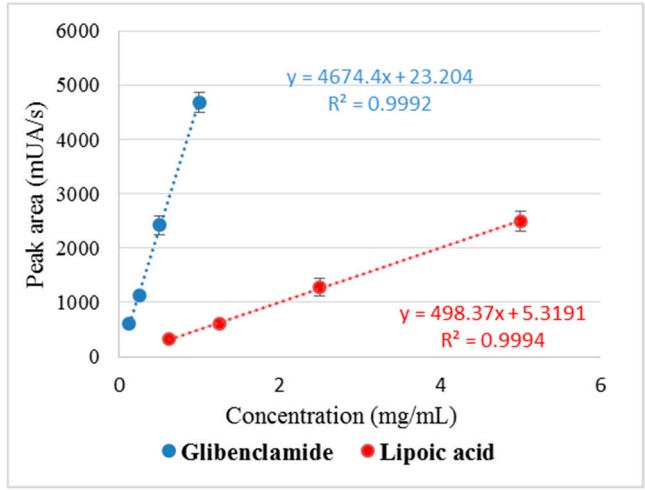

**Figure 3.** Standard curve and correlation coefficient ($R^2$) values for glibenclamide and lipoic acid.

Entrapment efficiency (EE%) and drug loading (DL%) were calculated based on the amount of active substance (Gly, LA) unloaded, which remained in TPP solution after the preparation of microparticles, using a validated HPLC method. The results obtained at the study of entrapment efficiency and drug loading are presented in Figures 4 and 5.

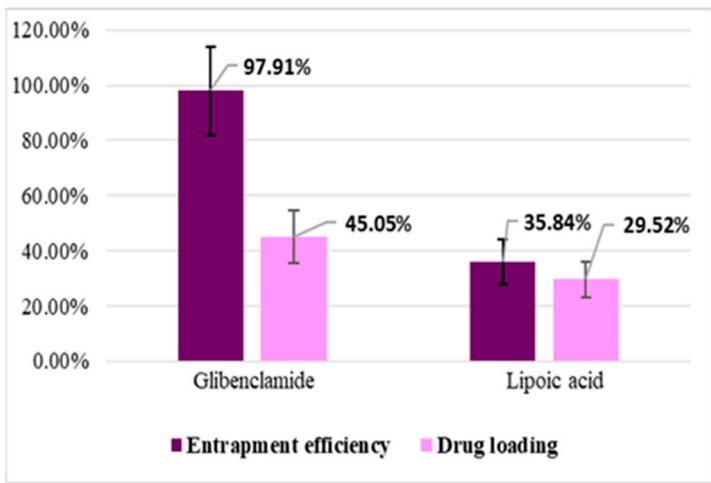

**Figure 4.** Entrapment efficiency (EE%) and drug loading (DL%) for CS–Gly and CS–LA systems.

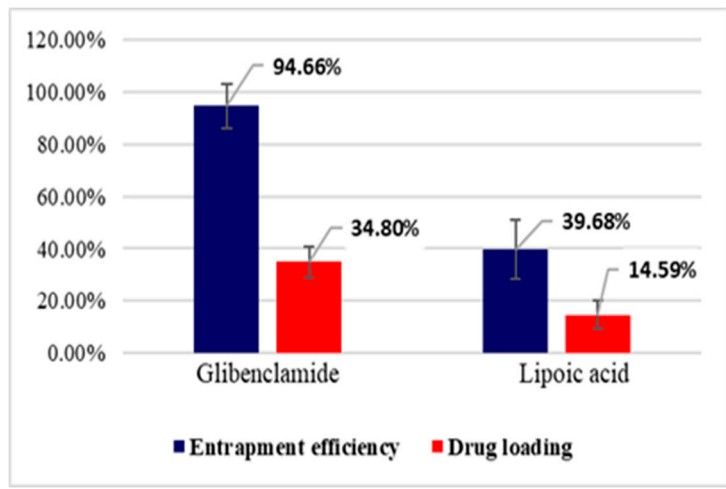

**Figure 5.** EE% and DL% for the systems type CS–Gly–LA systems.

### 3.4. Swelling Degree

SD% of the chitosan (CS), chitosan-glibenclamide (CS–Gly), chitosan-lipoic acid (CS–LA), and chitosan–glibenclamide–lipoic acid (CS–Gly–LA) microparticles in distilled water and SGF are presented in Figures 6 and 7.

### 3.5. Kinetic Release Studies

The release of the active substances was studied in conditions that simulate the physiological media, SGF (pH = 1.6) for 2 h and SIF (pH = 6.5) for 8 h, respectively, in appropriate conditions in terms of temperature and stirring.

For CS–Gly and CS–LA systems (Figure 8), the kinetic release of LA evidenced an increased release in the first 300 min, followed by a slower rate of the process in the interval 300–480 min, then the releasing percent being constant, 84.79%. For Gly, the absence of the release in the first 60 minit can be noticed, followed by an accelerated rate in the interval 60–120 min, followed by a slower rate of the process. The releasing percent was 52.49%, a fact that could be explained based on its lipophilic character.

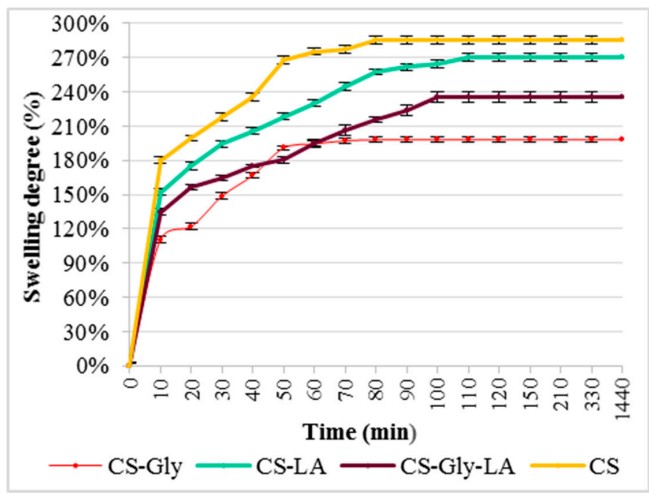

**Figure 6.** Swelling degree (%) in distilled water.

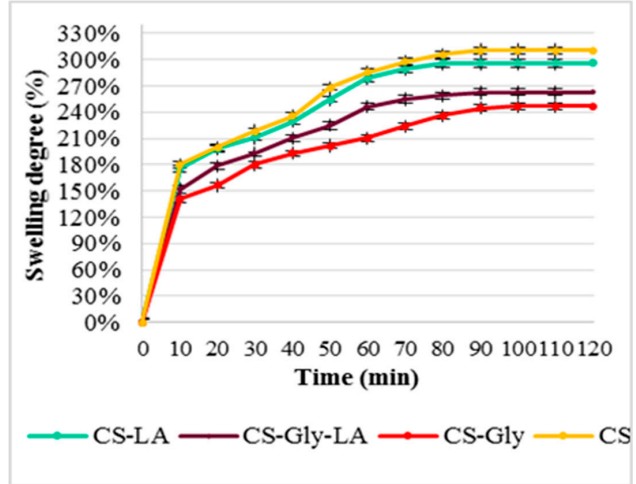

**Figure 7.** Swelling degree (%) in simulated gastric fluid (SGF).

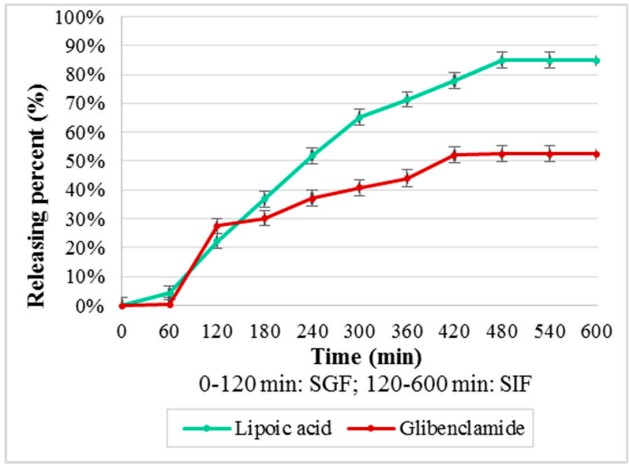

**Figure 8.** Releasing percent (%) from CS–Gly and CS–LA systems.

Concerning CS–Gly–LA systems (Figure 9), the release of LA started quickly and increased progressively during the first 300 min of the experiment, then the releasing percent being approximatively constant. For Gly, the release was slower, being insignificant during the first

240 min and increasing after 300 min. The releasing percent was higher for LA (88.68%) than for Gly (75.17%), a fact explained, also, through the difference of hydrophilic–lipophilic balance.

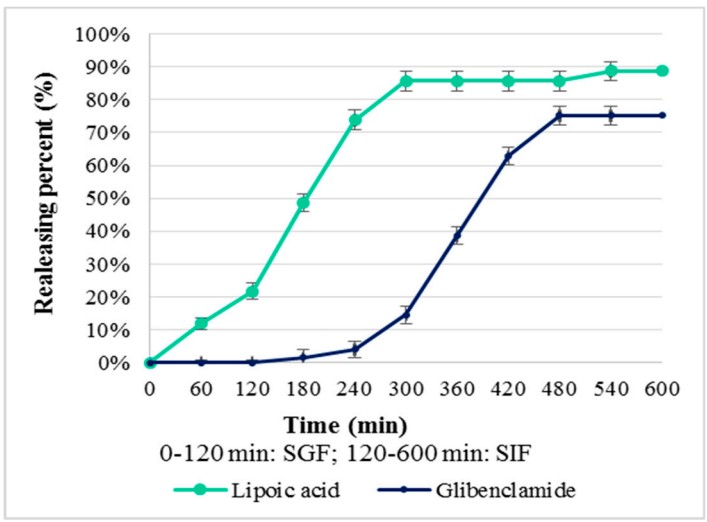

**Figure 9.** Releasing percent (%) from CS–Gly–LA system.

## 4. Discussion

Glibenclamide (Gly) and lipoic acid (LA)are often co-administered as separate pharmaceutical forms, this association targeting lowering blood glucose levels and managing diabetic neuropathy, together with other benefits. Consequently, the development of innovative multi-components dosage forms presents a significant interest.

New polymeric system-type microparticles based on chitosan (CS) were obtained, in which Gly and LA were included in the matrix.

The analysis of the results evidenced that EE% and DL% of the active substances (Gly, LA) into the CS matrix are dependent on the characteristic of the active substances in terms of molecular weight and hydrophilic–lipophilic balance. For LA, which has a pronounced hydrophilic character and low molecular weight, EE% was 35.84% for the CS–LA microparticles, and 39.68% for the CS–Gly–LA microparticles.Related to Gly, a substance with pronounced lipophilic character and higher molecular weight, EE% was significantly higher, 97.91% for the CS–Gly microparticles, and 94.66% for CS–Gly–LA microparticles, respectively. So, it could be appreciated that the low molecular weight and hydrophilic character facilitate the release of LA from the polymer matrix, while the higher molecular weight and lipophilic character favor the retention of Gly into the polymer matrix. In addition to this, the favorable influence of the association of Gly and LA in the same polymeric matrix on the entrapment efficiency of LAcan be noticed; the EE% of LA increased from 35.84% in CS–LA microparticles to 39.68% in CS–Gly–LA microparticles. At the same time, the EE% of Gly decreased insignificantly, from 97.91% to 94.66%. These data support the benefit of including both active substances into the CS matrix, CS–Gly–LA microparticles presenting the theoretical premises for a multi-target treatment of diabetes mellitus.

Pertainingto the swelling degree (SD%), the results highlight that the encapsulation of the active substances (Gly, LA) into the polymeric matrix was associated with decreasing of the swelling degree, both in distilled water and in SGF. It was also noted the SD% in SGF was higher than in distilled water, and the thermodynamic equilibrium was reached earlier (after 90 min, compared to 120 min for distilled water). These data support a better behavior in SGF than in distilled water, which could be beneficial for microparticle behavior in physiological gastrointestinal media.

The polymeric systems were formulated for oral administration, this route being more acceptable for patients. The kinetic release of the active substances is an important step in the characterization

of the polymeric systems, being a landmark for in vivo behavior of the systems developed. From the obtained results (Figures 8 and 9), the releasing percent as function of time, it can be observed that the release of the active substances from the polymeric matrix followed a different kinetic, depending on their hydrophilic–lipophilic balance. The results reveal the favorable influence of the LA on the release of Gly from the chitosan matrix, the releasing percent for Gly being 52.49% from CS–Gly microparticles, and 75.17% from CS–Gly–LA microparticles.

## 5. Conclusions

Diabetes mellitus is a chronic metabolic syndrome with great impact on the public health system due its increased prevalence in the last years. This disorder is caused by deficiency of insulin secretion, damage of pancreatic beta-cells, or peripheral tissue insulin resistance. Despite the promising antidiabetic agents, the major challenges of diabetes mellitus treatment include optimizing the currently therapies in order to achieve an optimum and balanced glucose level, as well as reducing associated diabetes complications. This paper presents the design and development of new chitosan-based polymeric systems containing glibenclamide and lipoic acid (CS–Gly–LA), as a new multi-target system with improved antidiabetic effects. Gly is a sulfonylurea with a stimulating insulin secretion effect, and LA is known mainly for its antioxidant effect. To these, the antidiabetic and antioxidant effects of chitosan (CS) are added. The encapsulation of active substances into CS matrix was proven through the presence of the absorption bands corresponding to the specific groups of Gly and LA. SEM microscopy revealed a spherical shape with a slightly rough surface and the recorded values for EE% and DL% revealed the importance of the features of the active substances, in terms of molecular weight and hydrophilic–lipophilic balance (for the entrapment process). The swelling data showed a better behavior of CS–Gly–LA in SGF, which means a favorable behavior in physiological gastrointestinal media, and support it as a proper system for oral administration. In addition, the kinetic release data support the good influence of the LA on the release of Gly from the chitosan matrix. The physicochemical and kinetic data of CS–Gly–LA polymeric system revealed the potential as a multi-target system and recommend it to be tested on experimental induced diabetes mellitus.

**Author Contributions:** Conceptualization, L.-G.C. and L.P.; methodology, A.B. and L.P.; software, S.C. and M.I.C.; validation, I.M. and L.P.; investigation, L.-G.C., S.C., and C.G.T.; writing—original draft preparation, L.-G.C. and M.I.C.; writing—review and editing, I.M. and L.P.; supervision, L.P.; project administration, L.-G.C. All authors have read and agreed to the published version of the manuscript.

**Funding:** This research activity was financially supported by L'ORÉAL-UNESCO through the fellowship program "For women in science", AUF-IFA 2019-2020, contract no. 28/2019and the APC was partially funded by "Grigore T. Popa" University of Medicine and Pharmacy in Iasi, Romania.

**Conflicts of Interest:** The authors declare no conflict of interest. The funders had no role in the design of the study; in the collection, analyses, or interpretation of data; in the writing of the manuscript, or in the decision to publish the results.

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
