# Peer review of "Chitosan-Based Delivery Systems Loaded with Glibenclamide and Lipoic Acid: Formulation, Characterization, and Kinetic Release Studies"

_applsci, doi:10.3390/app10217532_

Round 1

Reviewer 1 Report

The article is devoted to design of polymeric systems containing glibenclamide and lipoic acid for treatment of diabetes mellitus (oral administration). The study is clearly presented and showed promising results. To improve the article I suggest to add standard deviations (±SD) to the data presented on Figures 3-9.

Author Response

Dear Reviewer,

Thank you for the evaluation of our article, as well as for the comments and recommendations. For the answer to comments and recommendations please see the attached document.

Thank you!

Reviewer 2 Report

The manuscript Chitosan-based selivery systems soaded with  glibenclamide and lipoic acid: formulation,  characterization and in vitro drug release studies submitted to Applied Sciences - Manuscript Number: applsci966613 describes the synthesis and characterization of new polymeric systems based on chitosan (CS) in whose matrix were encapsulated glibenclamide (Gly) and lipoic acid (LA).

Some remarks have to be taken into account by the authors:

  1. In sec. 2.2, lines 85-87: the authors wrote that: “The mixture was stirred for 2 h, at 350 rpm; after that, the microparticles were obtained by adding, dropwise, 3 ml of the mixture obtained into 20 ml of TPP solution, under stirring (250 rpm) [20].” What type of device is used to dropwise of the mixture e.g. syringe or microsyringe or other device? What size is a drop? Sufficient details of preparation of polymeric systems, the microparticles and techniques should be described to allow others to replicate and build on published results.
  2. In Fig. 2, the legend is not clear. Please correct.
  3. In Fig. 2, there is no result of pure chitosan. In my opinion, these result should be added and compared with other
  4. In Figs. 4 -8, error bars should be included. Please correct.
  5. How is the reproducibility of the microparticles? This should be evidenced by the repeating of the preparation experiments and characterization and evaluation of the results.

Author Response

(The authors gave the same response as above.)

Reviewer 3 Report

This manuscript describes Chitosan-based delivery systems of glibenclamide and lipoic acid for the management of diabetes miellitus. Despite some experimental results are quite interesting, the manuscript does not reach the minimum quality for a scientific article to be published in the present form. However, I would consider to supports its publication after major revisions of the main manuscript.

  • The introduction of the manuscript is astonishingly short, confusing and do not provide a context to introduce the following scientific results. In this case, an adequate introduction must respond these questions:
    • What is the main issue treating diabetes miellitus?
    • Why the actual treatments do not provide a solution and must be improved?
    • Why CS/Gly/La drug release vehicles represent and innovation in this field?  
  • Figure 1 presents the SEM images of polymeric microparticles of the different polymeric systems:
    • The methodology of obtention of the images must be improved. There is any treatment performed on the organic surface to enhance electron conductivity?
    • Scale bar must be visible in all images.
    • Labelling system of figure 1 leads to confusion. The labeling is not use for the discussion of the Figure 1.

  • Figure 2 presents the IR spectra of glibenclamide (Gly), lipoic acid (LA) and CS-Gly-LA microparticles.
    • For the discussion is mandatory to include the IR spectre of CS. Otherwise, IR spectra do not provide enough evidence of the successful formation of CS-GLy-LA microparticles.
    • The presented spectra are overlapped among them and with the graph legend. This must be correct.
    • Spectre of Gly present low-resolution. This may indicate a problem during the analysis or there was low concentration of Gly during the performance of the IR. I would recommend to the authors to repeat this spectre.
    • The discussion of the Figure 2 does not convince me that CS-Gly-LA was obtained. Despite the presence of some characteristic bands in CS-Gly-LA, the spectre looks really different in comparison with the others. I suppose that the inclusion of CS spectre would provide enough evidence for the discussion of Figure 2.
  • Figure 6 and figure 7 figure 8 and figure 9
    • The manuscript is confusing in why are necessary to study the swelling degree of the polymeric siystems.
    •  I would need error bars in all of these graphics, especially for the swelling degree. The differences presented in figure 6 and 7 between the studied samples are small and could be attributed to methodology error.

  • The conclusions are extremely short and do no provide valuable information.

Author Response

(The authors gave the same response as above.)

Reviewer 4 Report

In this manuscript, the authors co-encapsulated glibenclamide and lipoic acid into a chitosan-based delivery system with a crosslinker pentasodium tripolyphosphate. The particle size, morphology, drug loading efficiency, swelling degree and drug release were characterized. There are some concerns that the authors need to be addressed.

  1. Please measure the FTIR spectrum of blank chitosan particles and compare with the difference between blank CS and CS-Gly-LA FTIR spectrum in figure 2.
  2. There is no error bar in figure 3, please add the error bar of the standard curve. Five data points or more are usually used to plot a standard curve. If concentration ranges of glibenclamide and lipoic acid can cover the HPLC reading of the samples, the standard curve can be used, but if not, please add more data points of higher or lower concentration to plot the standard curve.
  3. The reading and original spectrum of HPLC for sample should be listed in supplementary data document.
  4. Please correct “,” to “.” In figure 4.
  5. The original data of W1 and W2 of each particle formulation should be provided in supplementary data document. There is no error bar in figure 6 and 7, if the sample were tested triplicate please add the error bar.
  6. Please add the error bar in figure 8 and 9. Each formulation should be measured triplicate.

Author Response

(The authors gave the same response as above.)

Reviewer 5 Report

The paper entitled "Chitosan-based delivery systems loaded with Glibenclamide and Lipoic acid: formulation, characterization and in-vitro drug release studies" has been reviewed. The present work is appreciable, the design of microparticles and their characterization is appropriate and the results are promising.

However, before the manuscript might be suitable for publication, the authors should consider the following comments and include the following revisions:

1) Could you add in the legend of figures 8 and 9 the buffer that it has been used for kinetic release studies (SGF or SIF)? I presume that the two figures are referred to drug release in SIF. Is it possible to add also the data regarding kinetic release studies carried out in SGF?

2) Is it possible to include at least a chromatogram of CS-Gly-LA system obtained during kinetic release studies at any timepoint.

3) To simulate the physiological conditions, it would be important to understand the behaviour of developed mixed system (CS-Gly-LA) in presence of enzymatic pathways. Consequently, it strongly advises to measure the releasing percent of Gly or LA or both in human serum or fetal bovine serum. 

4) "In vitro" usually refers to studies of biological properties in isolated systems such as specific cells. Hence, it is suggested to remove "in vitro" and leave only "kinetic release studies". The title could be changed in "Chitosan-based delivery systems loaded with Glibenclamide and Lipoic acid: formulation and physicochemical characterization. In fact, the physicochemical characterization includes kinetic release studies, drug loading, swelling degree and etc.

5) It advises to replace "ml" with "mL" in the whole manuscript.

Author Response

(The authors gave the same response as above.)

Round 2

Reviewer 4 Report

The revised version addressed all my concerns. The manuscript can be accepted.

Reviewer 5 Report

The quality of the manuscript has been improved by the authors who have taken into consideration almost all weaknesses highlighted by the reviewers; the present paper is now suitable for the publication in Applied Sciences.